

# A method for high precision sequencing of near full-length 16S rRNA genes on an Illumina MiSeq

Catherine M. Burke and Aaron E. Darling

The i3 Institute, University of Technology Sydney, Sydney, NSW, Australia

## ABSTRACT

**Background:** The bacterial 16S rRNA gene has historically been used in defining bacterial taxonomy and phylogeny. However, there are currently no high-throughput methods to sequence full-length 16S rRNA genes present in a sample with precision.

**Results:** We describe a method for sequencing near full-length 16S rRNA gene amplicons using the high throughput Illumina MiSeq platform and test it using DNA from human skin swab samples. Proof of principle of the approach is demonstrated, with the generation of 1,604 sequences greater than 1,300 nt from a single Nano MiSeq run, with accuracy estimated to be 100-fold higher than standard Illumina reads. The reads were chimera filtered using information from a single molecule dual tagging scheme that boosts the signal available for chimera detection.

**Conclusions:** This method could be scaled up to generate many thousands of sequences per MiSeq run and could be applied to other sequencing platforms. This has great potential for populating databases with high quality, near full-length 16S rRNA gene sequences from under-represented taxa and environments and facilitates analyses of microbial communities at higher resolution.

## INTRODUCTION

Amplifying and sequencing 16S rRNA genes from microbial communities has become a standard technique to survey and compare communities across space, time and environments. High-throughput sequencing methods have made bacterial community profiling routine and affordable. However, this has come at the expense of read length with most platforms covering 250–600 bp of the ~1,500 bp 16S rRNA gene, where increases in read length are generally accompanied by decreases in read accuracy. Depending on the region sequenced, shorter fragments give variable taxonomic and phylogenetic resolution (*Claesson et al., 2010*; *Ghyselinck et al., 2013*; *Schloss, 2010*) and fail to resolve differences outside the targeted region, which may be ecologically relevant (*Denef et al., 2010*; *Fitz-Gibbon et al., 2013*; *Moore, Rocap & Chisholm, 1998*).

The rise of the shorter fragment, high-throughput methods has also resulted in a lack of quality, full-length reference sequences being deposited into reference databases, limiting our ability to classify shorter reads from taxa that are underrepresented in these

Corresponding author
Catherine M. Burke,
Catherine.Burke@uts.edu.au

databases (*Schloss et al., 2016*). Interpretation of 16S rRNA gene amplicon sequencing data is further confounded by PCR and sequencing artefacts including chimeras, biased amplification and sequencing errors. Some of these artefacts can be removed computationally (*Schloss, Gevers & Westcott, 2011*), but nevertheless lead to errors that artificially inflate diversity estimates (*Faith et al., 2013*; *Kunin et al., 2010*; *Lundberg et al., 2013*) and mislead analysis.

In this study, we have developed a method for sequencing near full-length 16S rRNA sequences on the high-throughput Illumina MiSeq platform. We provide proof of principle for the method by application to the skin microbiota with the reconstruction of high quality near full-length sequences. This method additionally provides the ability to remove putative chimeras and amplification bias.

## METHODS

### Extraction of microbial DNA from foot skin

DNA was extracted from skin swabs taken from the feet of three different healthy individuals. Twelve samples were taken in total. Skin swabs were collected by swabbing either the ball or heel area of the left or right foot with a rayon swab moistened in a solution of 0.15 M NaCl and 0.1% Tween 20. The swab was rubbed firmly over the skin for approximately 30 s. Swab heads were cut into bead beating tubes, and DNA was extracted from the swabs using the BioStic DNA extraction kit (Mo-Bio), as per the manufacturers instructions. DNA was quantified on a Qubit with a HS-DNA assay (Life Technologies). Ethics approval for this study was given by the University of Technology Sydney Human Research Ethics Committee (approval number 2013000170), and participants provided written consent.

### Preparation of short read 16S rRNA gene libraries for Illumina sequencing

Libraries (n = 12) of the V4 region of the 16S rRNA gene were prepared for Illumina sequencing from the microbial foot skin DNA samples using a modification of a previously published method (*Caporaso et al., 2012*). Briefly, samples were amplified using primers based on the Caporaso design (*Caporaso et al., 2012*), which were modified to include 8 nt rather than 12 nt barcodes and include a barcode on both the forward and reverse primer. The V4 region was amplified from 500 pg template DNA using 10 cycles of PCR with the modified Caporaso primers (V4_forward and V4_reverse), using different barcoded primer pairs for each sample (Table S1). After removal of excess primer via a magnetic bead clean up the samples were pooled and subjected to a further 20 cycles of PCR to enrich for amplicons containing the Illumina adapters, using primers Illumina_E_1 and Illumina_E_2 (Table S1). Pooling of samples during the enrichment PCR allows for an assessment of the putative recombination rate, by examining the rate of invalid barcode combinations that occur in the final paired end sequencing data. The method for each PCR reaction is described in detail below.

PCRs were carried out with a Taq core PCR kit (Qiagen), under the following conditions. For the initial 10 cycle PCR, reactions contained 1 × PCR buffer, 1 × Q

solution (Qiagen), 250 µM dNTPs, 0.5 µM each of V4_forward and V4_reverse barcoded primers, 500 pg DNA template, and 1.25 U Taq DNA polymerase in a 50 µl reaction volume. Thermal cycling was carried out at 95 °C for 2 min, followed by 10 cycles of 95 °C for 15 s, 50 °C for 30 s and 72 °C for 90 s, followed by a final extension at 72 °C for 5 min. After a magnetic bead clean-up using 0.8 volume of SPRIselect beads (Beckman Coulter), the cleaned PCR reactions were pooled and used as input for the second PCR reaction. This PCR contained 1 × PCR buffer, 1 × Q solution (Qiagen), 250 µM dNTPs, 0.25 µM each of Illumina_E_1 and Illumina_E_2 primers (see Table S1), 31 µl pooled PCR products from the first PCR, and 1.25 U Taq DNA polymerase in a 50 µl reaction volume. Thermal cycling was carried out at 95 °C for 2 min, followed by 20 cycles of 95 °C for 15 s, 55 °C for 30 s and 72 °C for 90 s, followed by a final extension at 72 °C for 5 min. These PCR reactions were again cleaned via a magnetic bead clean-up as above and run on an Agilent Bioanalyzer using a HS-DNA chip to confirm the amplicon size and determine the concentration. Negative control PCRs were included at all stages, and all PCR products were discarded if there was any evidence of a product in the negative controls.

The short read 16S rRNA libraries were sequenced using a Nano flow cell and a 500 cycle V2 kit on an Illumina MiSeq, using custom primers as described previously (*Caporaso et al., 2012*). This method will be referred to as "short sequencing" and data produced with this method as "V4" data. Read pairs were merged with FLASH (*Magoc & Salzberg, 2011*) and de-multiplexed using a new module implemented in a previously published version of PhyloSift (*Darling et al., 2014*).

## Sequencing of near full-length 16S rRNA gene sequences on the Illumina MiSeq

We present a method to sequence near full-length 16S rRNA gene amplicons using Illumina technology. The technique incorporates randomized molecular tags on both ends of individual 16S rRNA gene template molecules prior to PCR amplification. Copies of the templates are fragmented and sequenced and the dual tag information is used to accurately re-assemble near full-length 16S rRNA gene sequences. An overview of the method is shown in Fig. 1, and a detailed description is provided below.

### *Preparation of near full-length 16S rRNA gene libraries for Illumina sequencing with unique molecular tags on both ends*

Primers for amplification of the 16S rRNA gene contained the 27F (*Weisburg et al., 1991*) or 1391R (*Turner et al., 1999*) bacterial primer sequences, an 8 nt barcode sequence, a 10 nt random tag and partial Illumina PE adapter sequences (Fig. 2; Table S1). The use of a 10 nt random tag on both forward and reverse primers (~1 million possible unique tags at each end, ~1 trillion combinations) allowed us to uniquely tag each 16S rRNA gene molecule in our pool, by modifying previously described tagging approaches (*Faith et al., 2013*; *Lundberg et al., 2013*). Template DNA was subjected to one cycle of annealing and extension with the forward primer (long_forward, Table S1), followed by a magnetic bead clean up to remove excess primer, then another cycle of annealing and extension with the reverse primer (long_reverse, Table S1), followed by another magnetic bead clean up.

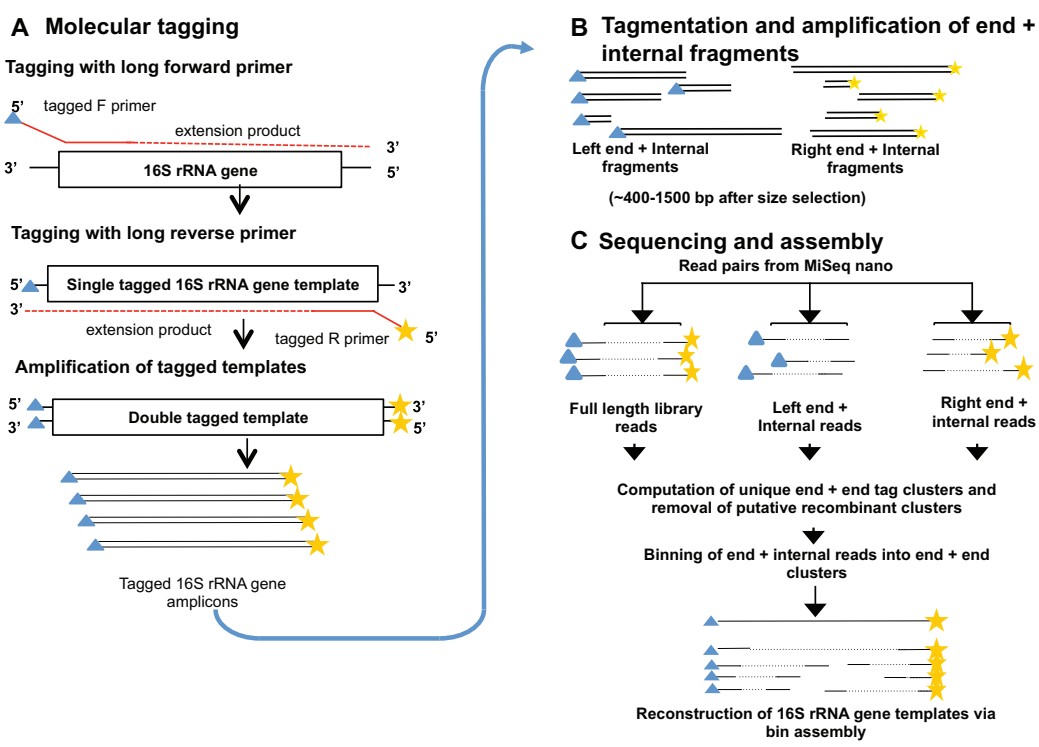

**Figure 1  Overview of the Long-16S method.** (A) 16S rRNA gene template molecules are tagged with unique tags via two single rounds of annealing and extension with tagged forward and reverse primers containing random tags (see Fig. 2), that also contain Illumina adapter sequences. After removal of tagged primers, tagged templates are amplified via PCR using primers complementary to the adapter sequences. Libraries from one or more samples can then be pooled and sequenced on the MiSeq. Blue triangles and yellow stars indicate random tags of 10 nt. (B) Full-length 16S rRNA gene amplicon Illumina libraries are tagmented using the standard Nextera method, and two pools of products are amplified which contain either the left end of the tagged amplicons and an internal region, or the right end of the amplicon and an internal region. This procedure adds Nextera adapters for sequencing at the internal end of the fragments. (C) Both full-length and tagmented libraries are paired end sequenced, and the unique molecular tags are used to computationally group sequences from the same progenitor 16S rRNA gene molecule for assembly of near full-length sequences.

The first PCR carries out extension of the 16S rRNA gene from the forward primer, which uniquely tags different 16S rRNA gene templates in the reaction. The second PCR uses extension products from the first PCR as a template to produce molecules with unique tags at both ends. While the original 16S rRNA gene molecules may also act as a template in the second PCR reaction, these products will only contain an Illumina adapter at one end and will therefore not be amplified in the enrichment PCR. The enrichment PCR (34 cycles) amplifies the tagged 16S rRNA gene molecule pool, using primers that are complementary to the Illumina adapter sequences at the ends of each tagged 16S rRNA gene molecule (primers PE_1 and PE_2, Table S1).

PCRs were carried out using the Taq PCR core kit (Qiagen), and differently barcoded primers were used for each sample. Reactions contained approximately 500 pg DNA template, 0.25 μM long_forward primer, 250 μM dNTPs, 1 × PCR buffer, 1 × Q solution, and 1.25 U Taq polymerase in a 50 μl volume. Cycle conditions were 95 °C for 1 min, 50 °C for 2 min then 72 °C for 3 min. This allows extension of the 16S rRNA gene

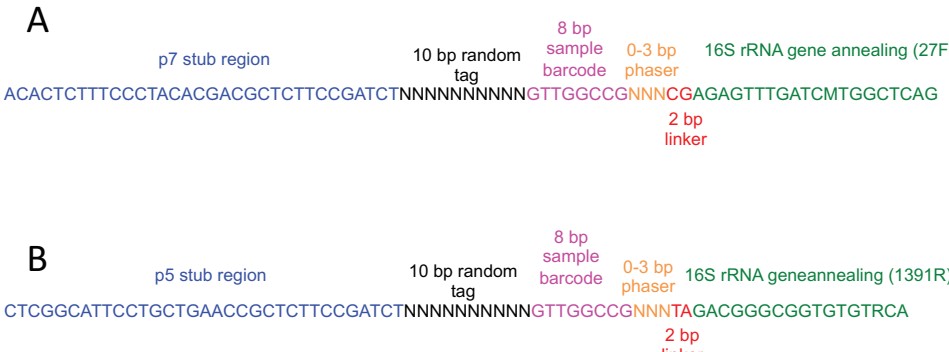

**Figure 2** Schematic of primers used for molecular tagging of 16S rRNA gene template molecules. (A) long_forward and (B) long_reverse. Stub regions correspond to Illumina adaptors for clustering on the MiSeq, and 0–3 nt phasers are included to increase nucleotide diversity between barcoded samples at individual sequencing cycles. 25 different barcodes were designed using software described in (*Meyer & Kircher, 2010*) for up to 625 different sample barcode combinations, which are listed in Table S1.

from the forward primer, which uniquely tags the forward end of each 16S rRNA gene molecule in the reaction. PCR reactions were then subjected to a magnetic bead clean up using 0.6 volumes of SPRIselect beads (Beckman Coulter) as per the manufacturer's instructions, except that the DNA was eluted in 35 μl nuclease free water. The second PCR was set up as described above, except that 0.25 μM of the long_reverse primer was used and the template was 31 μl of the bead-cleaned first round annealing and extension reaction. Only 31 of the 35 μl of bead cleaned first round PCR was used, to prevent contamination with the magnetic beads. Cycling conditions were as in the previous step: 95 °C for 1 min, 50 °C for 2 min and 72 °C for 3 min. During this second reaction, the uniquely tagged extension products from the first reaction act as the template to produce 16S rRNA gene molecules with unique tags on the forward and reverse ends. This was followed by another magnetic bead clean up, as described above, and the output of this step was used as a template for the final PCR reaction. The final enrichment PCR reaction contained 0.5 μM of each PE_1 and PE_2 primers, 250 μM dNTPs, 1 × PCR buffer, 1 × Q solution, 31 μl template (from the bead clean up) and 1.25 U Taq polymerase in a 50 μl volume. Cycling conditions were 95 °C for 2 min, followed by 34 cycles of 95 °C for 1 min, 58 °C for 30 s, 72 °C for 2 min, and a final extension of 72 °C for 5 min. PCRs were again subjected to a magnetic bead clean up as described above, before being analysed using a high-sensitivity DNA chip on a Bioanalyser (Agilent) to determine amplicon size and concentration. Negative control PCRs were included at all stages, and all PCR products were discarded if there was any evidence of a product in the negative controls. This was assessed via Bioanalyser traces from HS-DNA chips, although we acknowledge it is possible that products below the limit of detection may still have been present.

### Tagmentation of near full-length 16S rRNA gene amplicon libraries

The uniquely tagged, near full-length 16S rRNA gene PCR amplicons were subjected to tagmentation. The tagmentation procedure utilises a transposase to simultaneously

fragment the DNA while adding an adapter sequence for use on the Illumina platform. Tagmentation was carried out using the Nextera XT kit as per the manufacturer's instructions, with the exception of the PCR amplification step. Here, we split the tagmentation reaction into two and carried out two separate PCRs at half the volume specified in the kit (where normally only one PCR is carried out). Each PCR reaction contained a combination of one of the Illumina provided Nextera XT PCR primers and one of the primers from the enrichment PCR above, so as to amplify only those fragments of interest; specifically, we combined primers PE_1 and an Illumina Index 1 primer (N706) in one PCR reaction and PE_2 and an Illumina Index 2 primer (S504) in the second. We aimed to produce a pool of DNA fragments with either the PE_1 (forward end of the 16S rRNA gene amplicons) or PE_2 (reverse end of the 16S rRNA gene amplicons) sequences on one end and the i7 or i5 Illumina adapters (added to an internal region of the amplicon during the tagmentation reaction) at the other end, respectively. Each 25 µl PCR reaction contained 1 × Nextera PCR master mix with either 2.5 µl N706 Nextera index primer and PE_1 primer at 1 µM, or 2.5 µl S504 index primer and PE_2 primer at 1 µM, and 12.5 µl of the tagmentation reaction. PCR reactions were carried out as follows; 72 °C for 3 min, 95 °C for 30 s, then 12 cycles of 95 °C for 10 s, 60 °C for 30 s and 72 °C for 1 min, followed by a final extension at 72 °C for 5 min. These two PCR reactions provided a pool of fragments from across the 16S rRNA gene, which along with the full-length amplicons, could be paired-end sequenced on the MiSeq. PCR products from the tagmentation reaction were cleaned using 0.6 volumes of SPRIselect beads according the manufacturer's instructions, to remove fragments smaller than 400 bp. This step was necessary to achieve the desired range of fragment sizes (~400–1,500 bp) to ensure adequate coverage across the full span of the 16S rRNA gene amplicon.

### Sequencing of near full-length and tagmented 16S rRNA gene amplicon libraries

The molarity of both near full-length and tagmented 16S rRNA gene amplicon libraries was measured via an Agilent Bioanalyser High Sensitivity DNA chip. For the tagmented libraries, molarity was calculated based on the 400–1,000 bp range. Tagmented libraries were pooled at equal molarity and combined with the full-length amplicons at a molar ratio of ~7:1 (tagmentation pool: full length amplicons), with the tagmented pool at ~2.6 pM for loading. The pool was sequenced with 2 × 250 paired end reads, on a MiSeq Nano flow cell.

### Reconstructing full-length 16S rRNA gene sequences from tagged Illumina reads

Sequencing produced data from two kinds of fragments, those that span the entire 16S rRNA gene (end+end fragments) and those that contained either the forward or reverse end of the gene at one end with a region in the middle of the gene at the other end (end+internal fragments). Sequences from end+end fragments encoded a pairing of molecular tags and sample barcodes. Sequences were assigned to bins of original

16S rRNA gene progenitor molecules via the unique tags at either end of the molecule and were re-assembled to provide near full-length 16S rRNA gene sequences. Figure 3 shows an overview of the process.

To assign sequences to samples, the two 8 nt sample barcode regions were matched against the collection of known sample barcodes with up to one mismatch tolerated in each 8 nt barcode. Because internal regions of the 16S rRNA gene sequence might match a sample barcode, all reads with a potential sample barcode match were then screened for the presence of the proximal or distal 16S rRNA gene primer annealing sequence downstream from the sample barcode. Reads lacking a known sample barcode or the primer annealing sequence in one end were presumed to derive from an end+internal fragment.

*Consensus molecular tags and elimination of recombinants*
Due to sequencing error, the reads derived from the same template molecule may have had slightly different 10 nt random tagging sequences. To estimate the original 10 nt random tag sequences of tagged template molecules the UCLUST (*Edgar, 2010*; *Edgar, 2013*) algorithm was applied to identify clusters of matching random tag sequences at > 89% identity (e.g. one out of 10 bases mismatch) and to report the consensus sequences of these clusters. Clusters of molecular tags in the end+end fragments (the clustered sequences consisting of both 10 nt random tags, both 8 nt sample barcodes, and the first 14 nt of the 16S rRNA gene amplicon sequence in each read) were first identified. This was followed by the identification of the highest abundance cluster with the same combination of 10 nt random tags (one from either end) and discarding of any cluster containing one or both 10 nt random tags that were found in a different, more abundant cluster. This step aimed to identify and discard combinations of molecular tags that arose due to in vitro recombination. Recombinant forms are likely to be at lower abundance than the parental templates.

Finally, molecular tags from the entire set of reads (end+end and end+internal) were matched against the collection of consensus sequences and the reads were grouped into clusters for later assembly.

*Assembly of read clusters*
Each read cluster contained reads that, with high probability, originated from the same template molecule. A de novo assembly algorithm was applied on the read cluster to reconstruct as much of the original template molecule as possible. The reads were assembled using a version of the A5 pipeline called A5-miseq (*Coil, Jospin & Darling, 2015*) that has been modified to support assembly of reads up to 500 nt long and to trim out adapter sequence from reads instead of discarding reads containing adapter sequence. Only the first two stages of the A5-miseq pipeline were applied, involving adapter trimming, quality trimming, error correction, and contig assembly.

This method of amplifying, sequencing and assembling 16S rRNA gene sequences will be referred to as "Long-16S" and data produced with this method as "Long-16S" data, from here on in.

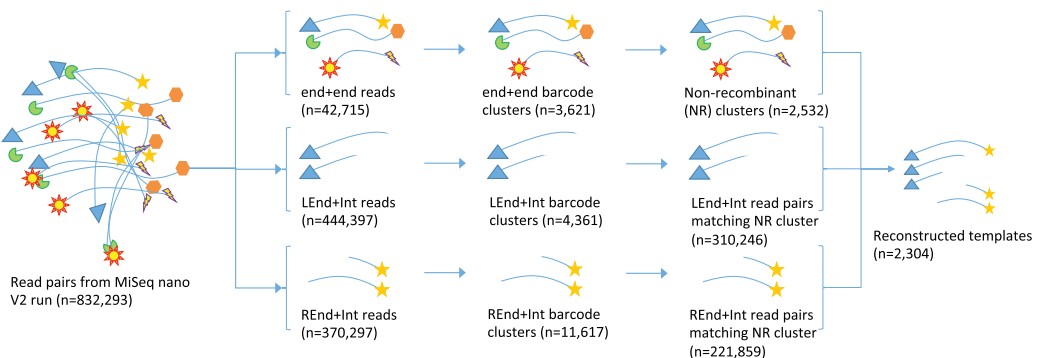

**Figure 3 Schematic demonstrating the processing of read pairs from the MiSeq to reconstruct near full-length 16S rRNA gene sequences.** Read pairs are placed into groups of end+end sequences, or end+internal sequences. End+end sequences are clustered into groups containing the same combination of random molecular tags from the two ends and putative recombinant clusters are removed (identified as having one or both molecular tags from a separate, more abundant cluster). End+internal sequences are assigned to clusters based on their unique molecular tags, and each cluster is used to generate an assembly of the full-length sequence.

### Assessment of assembled Long-16S sequence quality

The accuracy of the base calls was estimated by calculating PHRED scale quality scores (*Ewing et al., 1998*) using samtools (*Li et al., 2009*). Briefly, the reads present in each assembled molecular tag cluster were mapped back to the assembled contigs using BWA MEM (*Li, 2013*). From the mapped reads, a consensus FastQ sequence was called using samtools, bcftools, and vcfutils.pl (*Li et al., 2009*). The quality scores in the resulting FastQ file were then used for subsequent quality analysis and visualization.

### Removal of chimeras in cluster assemblies

Putative chimeras were identified in end+end reads as described above; this permitted estimation of the overall recombination rate and the frequency of recombinant fragments relative to full-length fragments for each cluster. However, it was not possible to directly identify end+internal reads derived from a chimeric fragment using molecular tags, as some of these reads contained a molecular tag that matched an original template cluster. Erroneous signal from these reads was eliminated in two ways, both of which depended on reads derived from the recombinant form existing at lower abundance in the sequence data. First, during the initial assembly process, k-mer error correction and consensus generation eliminated differences in the sequence present in low abundance chimeric reads. Second, in cases where the cluster assembly contained multiple contigs, the depth of coverage of contigs was used to identify and remove contigs at much lower abundance than the dominant contigs in the cluster. For the present work, we removed any contigs with an average coverage that was 10-fold lower than that of the highest abundance contig. Future work could use information derived from the end+end sequences to estimate the expected fraction of recombinant reads in a cluster and use this to aid the process of eliminating chimera-derived contigs or to identify clusters for which recombinant elimination may not be possible.

## Analysis of V4 and Long-16S data

Both V4 and Long-16S data generated from the 12 skin samples were analysed using the software package QIIME (*Caporaso et al., 2010*). For comparison, the corresponding V4 region was extracted from the Long-16S sequences (which we will refer to as extracted-V4). Only those Long-16S sequences that were > 1,300 nt in length were included in the downstream analysis. V4 sequences were initially quality filtered using the default settings, with the exception of sequence length, which was altered to remove sequences less than 240 nt and longer than 260 nt. V4 sequences were additionally checked for the presence of chimeras using the UCHIME (*Edgar et al., 2011*) method, both against a reference database (RDP Classifier 16S training set No. 9, accessed from https://sourceforge.net/projects/rdp-classifier/files/RDP_Classifier_TrainingData/), as well as using the dataset itself as the reference. Long-16S sequences were quality filtered using default settings and excluding sequences longer than 1,400 nt. Quality filtered sequences from the V4, Long-16S and extracted-V4 datasets were then combined, and sequences were assigned to OTUs using the closed reference picking method, which assigns sequences to pre-clustered OTUs at 97% similarity from the chimera filtered Greengenes database (*DeSantis et al., 2006*; *McDonald et al., 2012*). OTUs with less than two sequences were filtered from the OTU table. Taxonomy was determined based on membership to the database of pre-clustered OTUs, and the relative abundance of taxa at different levels was generated using the *summarize_taxa.py* script. Differences in abundance of taxa at phylum and genera level between the V4 and Long-16S data were tested for significance using the *group_significance.py* script with taxa summary biom tables as input and using the Kruskal-Wallis test with the Benjamin Hochberg FDR correction for multiple testing.

## Assessment of bias reduction using unique molecular tags

The use of molecular tagging has previously been shown to reduce the effect of PCR bias in RNA-seq data, for better quantitative assessment of sequences from the original samples (*Islam et al., 2014*). Assuming that each uniquely tagged 16S rRNA gene molecule from our skin samples was present at the same abundance as all other uniquely tagged molecules (i.e. one copy of each) and that unbiased amplification would result in an equal abundance of each cluster, we estimated the amount of biased amplification that occurred during PCR by comparing the differences in the abundance of end+end sequence clusters.

# RESULTS

## Near full-length 16S rRNA gene sequences from an Illumina MiSeq generated by molecular tagging

Sequencing of both the full-length and the tagmented amplicon pools was successful with a cluster density of 400–500 k/mm$^2$ and 832,293 read pairs. Higher clustering and sequencing output may be possible by optimising the loading concentration and ratio of end+end and end+internal pools. Clustering of end+end reads resulted in 5,085 clusters. Of these, 2,265 (44.6%) were deemed to be putative recombinant clusters, with predicted parental templates on average 29 times more abundant than putative recombinants (Fig. 4). Putative recombinant end+end sequences represented 4,378 of the total 42,715

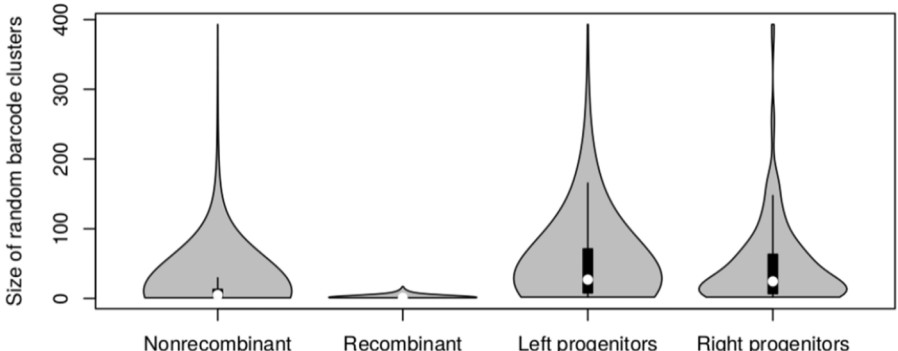

**Figure 4 Abundance of putative recombinants.** Violin plot showing the abundance of molecular tag clusters identified as putatively recombinant (left), along with abundances of the progenitor molecules producing recombinant forms. Parental templates were on average 29 times more abundant than the putatively recombinant forms. Median values are indicated by white dotes and the interquartile range by black boxes.

sequences in the end+end read pool, indicating an average recombination rate of 10.2% among all samples. After binning and assembly of end+end and end+internal read clusters, 2,304 16S rRNA gene sequences were assembled from 558,053 Illumina read pairs. Sequence lengths ranged from 449 to 1,372 nt (full-length), and ~70% of these were greater than 1,300 nt. The assembly of sequences with less than the expected length (i.e. those 400–1,300 bp) is possibly due to a lack of coverage across the internal regions for some end+end clusters. The range of sequence lengths generated is shown in Fig. 5.

Assembled sequences had consistently high quality scores across their length, with the per site average estimated PHRED quality scores at each position ranging from 54.0–89.5 (median 68.0) (Fig. 6A). This indicated estimated base-calling accuracies of greater than 99.999% at each position of the assembled 16S rRNA gene sequences. We note that errors due to base misincorporations that occur during early cycles of the enrichment PCR cannot be directly measured with this method, therefore these estimates of consensus accuracy may overestimate the true accuracy of the reconstructed 16S rRNA genes. Higher qualities for the Long-16S sequences were associated with higher coverage, which is particularly apparent at each end of the reconstructed sequences (up to 200 and beyond 1,200 nt), which were associated with one read from every read pair (end+internal or end + end) in the data set (Fig. 6B).

## Short sequencing of the 16S rRNA gene V4 region

A total of 296,864 paired end 16S rRNA gene V4 sequences were generated from the 12 skin samples. Of these sequences, 11,240 could not be assigned to a sample due to invalid forward and reverse barcode combinations (e.g. combinations which were never assigned to a sample), indicating an in-vitro recombination rate of at least 3.8%, although a small proportion of this may be due to cross-contamination of barcoded primers (*Kircher, Sawyer & Meyer, 2012*). These sequences were removed from the dataset. We note that in-vitro recombination could also create barcode combinations that would match a
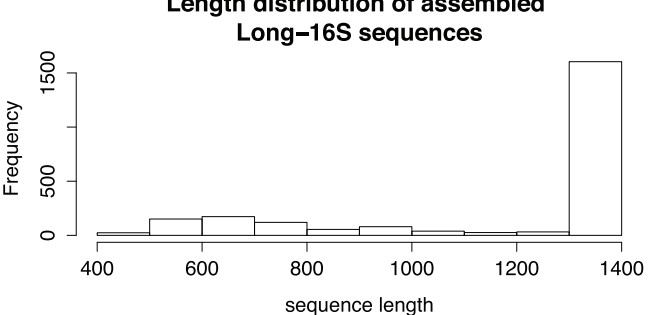

**Figure 5 The length distribution of assembled Long-16S sequences.** Sequence length ranged from 400 to 1,378 nt, corresponding to a full-length amplicon. 70% of the assembled sequences are > 1,300 nt in length.

**A**

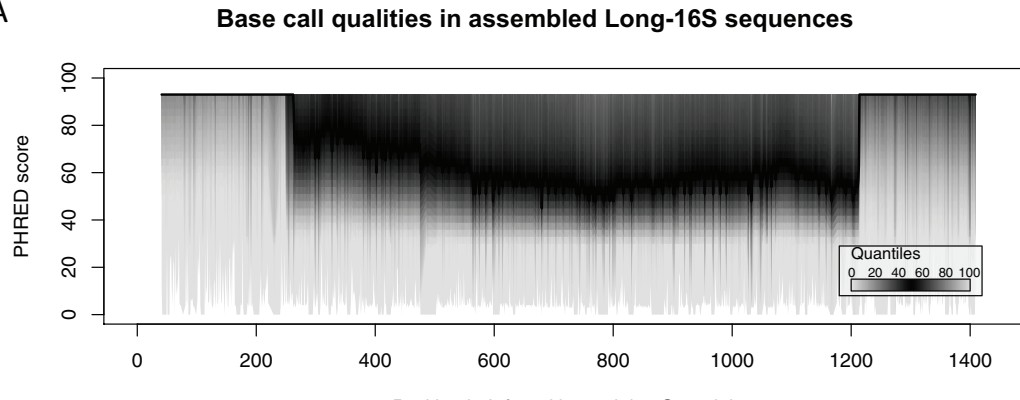

**B**

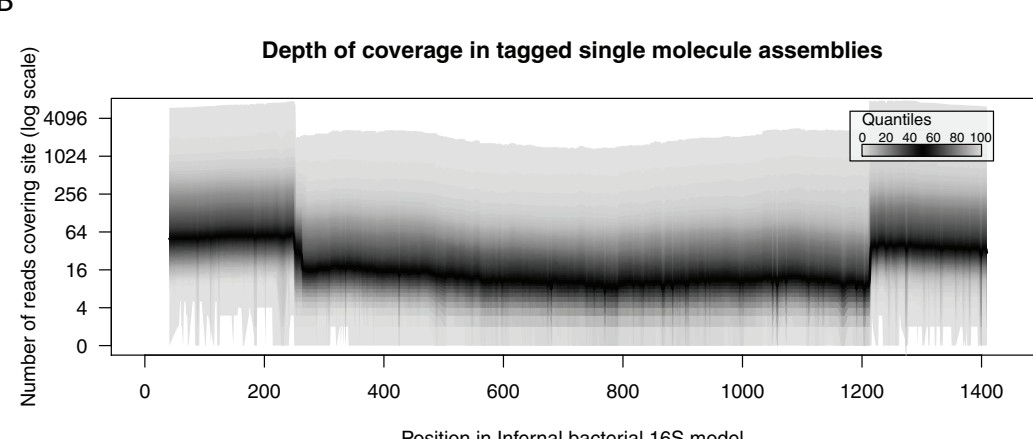

**Figure 6 Quality and coverage plots.** (A) Quality and (B) coverage plots showing median PHRED quality scores and associated coverage across the length of the Long-16S sequences. Higher qualities for the Long-16S sequences are associated with higher coverage, which is particularly apparent at each end of the amplicons (up to 200 and beyond 1,200 nt), which were associated with one read from every read pair in the data set.

valid sample and therefore be undetectable recombination events. In contrast, when attempting to detect recombination products using the chimera detection software UCHIME (as implemented in QIIME), only 0.05% of the sequences were flagged as

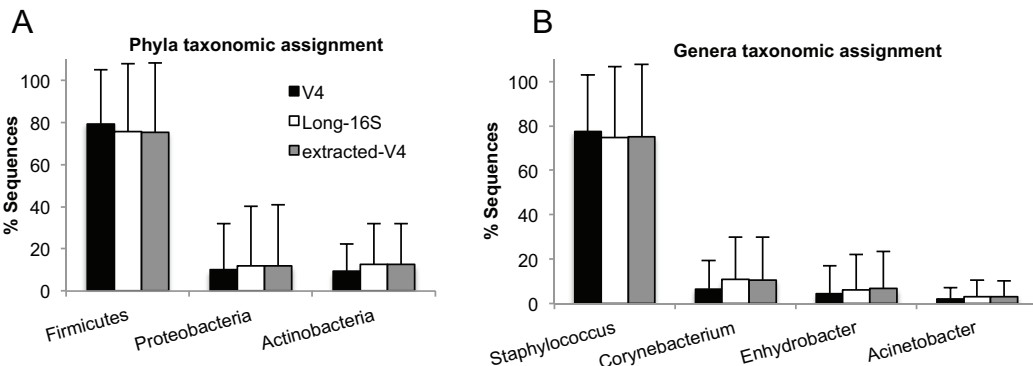

**Figure 7 Taxonomic assignments at the phylum and genus level.** The relative proportions of taxonomic assignments for (A) phyla and (B) genera are shown for OTUs from V4, Long-16S and extracted-V4 sequences (V4 region extracted from the Long-16S sequences). Similar taxonomic assignments are obtained across the V4 and Long-16S datasets. Only taxa comprising an average of more than 1% of sequences per sample are displayed. Bars represent the average value across 12 samples for each sequencing method, and error bars are the standard deviation.

chimeric when compared against a reference database (SILVA) and 0.2% when using the dataset itself as the reference. This highlights the difficulties of using software alone to detect recombination products from PCR in the absence of sample barcode and molecular tag information. Sequences that were flagged as chimeric using UCHIME, which had not been identified as chimeric based on sample barcode combinations (as described above) were also removed from the dataset.

## Assembled near full-length 16S rRNA gene sequences produce data consistent with short read sequencing

Taxonomy, as assigned in QIIME, was similar to previous reports for skin communities, dominated by Firmicutes, Actinobacteria, and Proteobacteria. Long-16S and extracted-V4 OTUs showed the same broad taxonomic distribution as the V4 sequence data (Fig. 7). There was a small decrease in the representation of Firmicutes and an increase in the representation of Actinobacteria and Proteobacteria (Fig. 7), however these differences were not significant (Kruskal-Wallis with Benjamin-Hochberg FDR correction for multiple testing, $p > 0.05$). Similar taxonomic assignments between the different sequencing methods were also observed at the level of genus, with communities dominated by *Staphylococcus*, followed by *Corynebacterium*, *Enhydrobacter* and *Acinetobacter*. The *Corynebacterium* genus had an increased representation in the full-length data set as compared to the V4 data, which likely accounts for the observed difference in representation for the Actinobacteria phyla, but as above, this difference was not significant (Kruskal-Wallis with Benjamin-Hochberg FDR correction for multiple testing, $p > 0.05$).

*Comparison at the OTU level*
Of the OTUs clustered at 97% similarity from the 12 libraries of Long-16S sequence data, an average of 22.7% ($\pm$ 15.6) were also found in matched sample V4 data that was

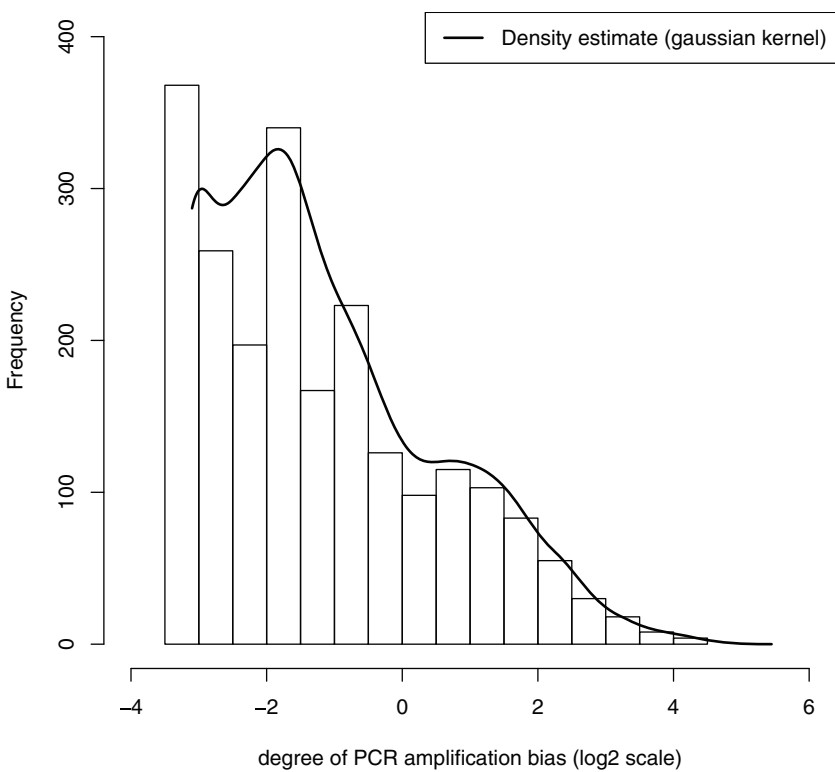

**PCR amplification bias**

degree of PCR amplification bias (log2 scale)

**Figure 8 Distribution of the estimated degree of PCR amplification bias.** Estimates of bias were calculated from the deviation of each end+end sequence cluster from the mean end+end sequence cluster abundance.

clustered in the same way. This disparity is likely due to comparing OTUs of sequences of different lengths and the way in which OTUs are defined in QIIME. Sequences are assigned to OTUs by clustering (UCLUST method) against a database of full-length representative sequences (from the Greengenes database (*DeSantis et al., 2006*)) which are at most 97% similar to each other and are used as centroids for each cluster. Sequences that are 97% similar across the full 16S rRNA gene may be more or less than 97% similar in the V4 region only, since different regions of the 16S rRNA gene evolve at different rates (*Schloss, 2010*). As such, V4 sequences will be assigned to the OTU cluster from the first representative sequence in the database that is at least 97% similar within the V4 region, while longer sequences that contain an identical V4 region but are not 97% similar to the same cluster centroid will be assigned to a different OTU. We therefore analysed OTUs clustered from the V4 region of the Long-16S sequences (extracted-V4 sequences) in comparison to OTUs clustered from the V4 data to determine whether we had captured similar OTUs with the V4 and Long-16S sequencing methods. In this case 83.7% (± 15.9) of extracted-V4 OTUs were shared with the matched sample V4 OTUs (Table S2). Although fewer sequences were present in the Long-16S data set, yielding many fewer OTUs overall, the data indicates that the newly developed method gives broadly congruent community profiles with respect to taxonomy and OTU clustering.

## Assessment of bias reduction using unique molecular tags

We estimated the amount of biased amplification that occurred during PCR by comparing the differences in the abundance of end+end sequence clusters. The average abundance was calculated from all clusters, and the relative mean error was 2.08 or 1.81 if singleton clusters (possible recombinants) were excluded. This indicates a standard deviation of approximately two times the average across the dataset under the particular amplification conditions used here. Figure 8 shows the distribution of the estimated amplification bias, which ranges from 0.06 to ~32 times the average cluster abundance. This potential bias is eliminated by considering each assembled sequence cluster as having a count of one.

## DISCUSSION

We have presented a method for sequencing near full-length 16S rRNA gene sequences on the high throughput Illumina MiSeq platform. This method utilizes tagging of individual 16S rRNA gene template molecules with unique, random sequences (tags), an approach that has been used previously to generate consensus sequences from short read data (*Faith et al., 2013*; *Lundberg et al., 2013*). These previous approaches targeted individual variable regions rather than sequencing of the whole gene, and incorporated tags at only one end of the amplicon (*Faith et al., 2013*) or tags too short (*Lundberg et al., 2013*) to permit the assembly and chimera filtering techniques proposed here. This new method incorporates randomized molecular tags on both ends of individual 16S rRNA gene template molecules prior to PCR amplification and uses this dual tag information to reconstruct near full-length 16S rRNA gene sequences and remove putative chimeras.

We assembled 2,304 16S rRNA gene sequences in a single MiSeq Nano run, 1,604 of which were longer than 1,300 bp. There are at least two factors that could contribute to the assembly of templates < 1,300 bp. The first is a lack of read coverage throughout the full sequence, causing fragmentation in the 16S rRNA gene assembly. A second likely contributing factor is collisions in molecular tags, e.g. when different templates receive the same molecular tag on one (or both) end by chance. Exact matches are expected to be rare when sampling ~5,000 items with replacement from a pool of 1 million (the number of random 10-mers). However, when mismatches are permitted in molecular tags to accommodate sequencing error, the expected rate of collisions increases. Distinguishing sequencing errors in molecular tags from tag collisions is an area for future research. Finally, a third factor that could contribute to these short reads is that although differences in the abundance of molecular tag pairs enable true full length templates to be distinguished from recombinant forms, our protocol does not provide direct information to distinguish whether the end+internal reads derive from a true template or a recombinant form. Therefore, the collection of end+internal reads associated with an end+end template may contain contaminating recombinant reads. When these contaminating reads are at sufficiently high abundance, and are sufficiently divergent from the other end+internal reads they can cause assembly fragmentation. Our data processing scripts (available in github) include steps to eliminate contigs that

have very low relative coverage within a molecular tag cluster, but the assembly fragmentation caused by the putative contaminant reads could remain.

While only a small amount of data were analysed here, we have provided proof of principle that this method is capable of generating many high quality, near full-length 16S rRNA gene sequences, using one of the most cost effective and widely available high-throughput sequencing platforms. Assuming linear scaling, the method could yield up to 80,000 full-length 16S rRNA gene sequences on a 600 cycle MiSeq v3 kit, while a HiSeq 2,500 might generate up to 480,000 near full-length 16S rRNA gene sequences in a single "rapid run" lane. This potentially places the reagent cost per 16S rRNA gene sequence in the region of US$0.006–$0.025, making the cost of producing these sequences much lower than traditional Sanger sequencing (~US$8 per sequence) (*Schloss et al., 2016*). Protocols for the generation of near full-length 16S rRNA gene sequences on other platforms have also been described (*Benítez-Páez, Portune & Sanz, 2016*; *Fichot & Norman, 2013*; *Schloss et al., 2016*; *Singer et al., 2016*), but at present they cannot match the high quality, throughput and cost efficiency of Illumina platforms.

This method is more expensive and has lower throughput than sequencing short regions of the 16S rRNA gene, but the cost could be justified where the increased resolution afforded by long 16S rRNA gene sequences is required. Accurate classification of short reads is dependent on the completeness of reference databases and training sequences used (*Werner et al., 2012*), and the high quality sequences generated with this method could be particularly useful for providing reference sequences for environments or taxa that are poorly represented in the current databases. Several recent studies utilized traditional cloning and Sanger sequencing of near full-length 16S rRNA genes in combination with high throughput short read sequencing for this purpose (*Chaves-Moreno et al., 2015*; *Dewhirst et al., 2015*; *Hund et al., 2015*). There is also growing interest in the ability to resolve species and strain level differences from microbiota data (*Eren et al., 2015*; *Greenblum, Carr & Borenstein, 2015*; *Kraal et al., 2014*; *Luo et al., 2015*; *Tikhonov, Leach & Wingreen, 2015*), and the additional information obtained with accurate near full-length 16S rRNA gene sequencing could be used to better identify putative strains of bacteria between and within samples.

We estimated a 100-fold reduction in average error rate with our method compared to paired-end sequencing of short regions. While estimated accuracy was very high, the presence of errors introduced by base misincorporation during PCR cannot be directly assessed in this dataset and will still be present. We chose to use a standard Taq polymerase over a high-fidelity polymerase, as preliminary experiments indicated an extremely high rate of recombination with the high-fidelity enzyme tested. Based on published estimates of the Taq polymerase error rate (~$3 \times 10^{-5}$) (*McInerney, Adams & Hadi, 2014*), we expect around one error per 10 tagged templates to occur prior to PCR enrichment. Errors that occurred during the enrichment PCR after tagging and after tagmentation were potentially corrected via the consensus sequence, depending on how early in the PCR individual errors occurred. The use of a mock community would allow for more robust testing of PCR and sequencing error rates, and other potential sources of error

common to all 16S rRNA gene amplicon sequencing protocols (e.g. PCR conditions and priming regions used).

Artifacts generated through the PCR process, including chimera formation and biased amplification of a subset of templates have been acknowledged to be a problem in surveys of microbial communities for some time (*Judo, Wedel & Wilson, 1998*; *Kopczynski, Bateson & Ward, 1994*; *Liesack, Weyland & Stackebrandt, 1991*; *Polz & Cavanaugh, 1998*). Despite attempts to minimize these effects via tuning of experimental parameters (*Fonseca et al., 2012*; *Judo, Wedel & Wilson, 1998*; *Smyth et al., 2010*) or computational detection (*Ashelford et al., 2005*; *Edgar et al., 2011*; *Haas et al., 2011*) these artifacts remain and may confound data analysis. The use of a dual tag system as demonstrated here offers an alternative signal for both the removal of putative chimeras, and the correction of PCR bias (*Islam et al., 2014*). Using this method, we were able to remove a large proportion of putative chimeric sequences from the dataset and estimate the degree of bias (Fig. 8). Because the abundance profiles of the reconstructed near full-length 16S rRNA gene sequences work on the assumption that each tagged template was originally present as a single copy, this method provides a way to minimise PCR bias when applied to microbial communities.

## CONCLUSION

We have provided proof of principle that this method enables the generation of large numbers of high quality, near full-length 16S rRNA gene sequences. We note that the method of dual molecular tagging could be applied to any sequencing platform and any amplicon target to enhance chimera removal and reduce amplification bias and base calling error. This is valuable for the expansion of current databases with high quality, near full-length reference sequences. Additionally, in conjunction with new algorithms (*Eren et al., 2015*; *Tikhonov, Leach & Wingreen, 2015*), this method could facilitate a finer understanding of population dynamics in microbial ecosystems.

## ACKNOWLEDGEMENTS

We thank Josh Quick for advice on configuring the MiSeq to cluster long fragments, Paul Worden for assistance operating the MiSeq instrument, and Torsten Thomas for helpful discussions and suggestions during manuscript preparation.

### Funding

This work was funded by internal grants from the University of Technology Sydney. The funders had no role in study design, data collection and analysis, decision to publish, or preparation of the manuscript.

### Competing Interests

The University of Technology Sydney has filed a provisional patent covering some aspects of the described work, under United Kingdom Patent Application No. 1409282.9, 23rd May 2014.

## Author Contributions

- Catherine M. Burke conceived and designed the experiments, performed the experiments, analyzed the data, contributed reagents/materials/analysis tools, wrote the paper, prepared figures and/or tables, reviewed drafts of the paper.
- Aaron E. Darling conceived and designed the experiments, performed the experiments, analyzed the data, contributed reagents/materials/analysis tools, wrote the paper, prepared figures and/or tables, reviewed drafts of the paper.

## Human Ethics

The following information was supplied relating to ethical approvals (i.e., approving body and any reference numbers):

UTS Human Research Ethics Committee.

UTS HREC REF NO. 2013000170.

## Patent Disclosures

The following patent dependencies were disclosed by the authors:

United Kingdom Patent Application No. 1409282.9, 23rd May 2014.

## DNA Deposition

The following information was supplied regarding the deposition of DNA sequences:

SRA accession number: SRX655489.

## Data Deposition

Software automating the process of sample demultiplexing, barcode clustering, and amplicon assembly is available on Github in the koadman/lonas repository:

http://github.com/koadman/longas.

## Supplemental Information

Supplemental information for this article can be found online at http://dx.doi.org/10.7717/peerj.2492#supplemental-information.

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
