# Peer review of "A method for high precision sequencing of near full-length 16S rRNA genes on an Illumina MiSeq"

_PeerJ, doi:10.7717/peerj.2492_

## Round 0.1 · original submission · Major Revisions

This manuscript presents a brilliant experimental design for generating a fairly large number of full-length 16S rRNA gene sequences from environmental samples by using a single sequencing platform.

The 3 reviewers have provided a large number of useful comments, which will improve the manuscript. Please pay extra attention to grammar and typos.

·

Basic reporting

With the brilliant experimental design, the study by Burke and Darling provides important means to generate a fairly large number of full-length 16S rRNA gene sequences from environmental samples by using a single sequencing platform.

The language of the manuscript is quite clear. Although, I stumbled upon multiple typos while reading it. I believe a careful examination of the text would be enough to clear all up.

Figures are pretty explanatory. However, there are some 'results' in figure captions (i.e., Figure 7, 8). Authors may want to consider keeping figure captions solely to describe the figure, and explain the details of findings in the text where the figure is referenced.

The use of space in Figure 1 could be improved to make a better use of the available space. When it appears in the final PDF, it's width will shrink a lot due to its length for it to fit in a single page.

Showing actual coverage values in log scale could be more intuitive in the Y-axis of Figure 6:B instead of showing the log of values.

I can't see the contribution of the phylogenetic analysis explained at Lines 282-291 to the study (at least in its current form). Cutting the V4 and V1-V4 regions of long reads and comparing their ability to resolve more branches on a common phylogeny is not relevant to what this study is about. Not only we already know that longer sequences should be able to better resolve branches, it shouldn't be a concern of this manuscript even if they didn't. In my opinion, removing any and every phylogenetic analysis along with Figure 8 from the manuscript would help the reader to focus on the truly important contributions of this study.

Experimental design

Molecular aspect: I am not a molecular biologists, therefore my reading of the preparation of short-read 16S libraries for Illumina sequencing was inevitably somewhat superficial. I am hoping that the other reviewer(s) will have more experience with lab techniques to critically review this part of the study. Yet, I can see that the insert size of Lend + internal and internal + Rend reads is critical to have almost the entire middle section of the reconstructed full-length 16S sequence to be evenly covered. I think Figure 1 (especially the C panel) could be improved to better show the random insert sizes to give that notion to the reader (maybe there could also be some partially overlapping pairs as examples right underneath the "Read pairs from MiSeq nano").

Computational aspect: Thanks to the brilliance of the molecular design, the computational needs to generate full-length sequences using the sequencing data is quite straightforward, and I think the authors did well in this aspect.

Validity of the findings

I find the discrepancy between the taxonomic profiles recovered via V4 and V4-long reads intriguing. Figure 7 shows an almost perfect concordance between the genus-level taxonomic assignments for long reads and V4-long, yet V4 reads differ. The authors point out that these differences are not significant (Line 344, Line 351), however, they may become significant in more complex datasets. Could this be a systematic bias where certain taxa (due to their sequence composition) is less likely to be assembled into full-length sequences? Although I am not suggesting more experiments for this study, I think the analysis of a mock community would have answered some of these questions. Maybe a line about this would suggest future directions for people who may want to do some follow up studies.

Additional comments

The Github link in the PDF leads to https://github.com/koadman/ithree/longas, which doesn't exist.

Line 44: Schloss PeerJ preprint 2015 is now published.

I find the 99.999% base-calling accuracy of each position of the assembled 16S sequences exciting (Line 317). Although, because the data is not coming from a known community, it is hard to really assess the true accuracy of the assembled reads, and assign a robust error rate for the approach. Thanks to the availability of the full-length 16S sequences, and the low complexity of the samples analyzed, I did a preliminary analysis to understand at what positions very similar sequences in the assembled data differ from each other (Here are the steps if the authors would like to reproduce my attempt: https://gist.github.com/meren/29e56db37d1cf385c98f; scripts that start with 'o-' and the entropy-analysis program are available in the oligotyping pipeline)). This analysis suggested many nucleotide positions with variation (https://imgur.com/yWsYXjg). Some of these reads in this group must be true sequences, but it is very likely not all of them are correct. I agree that the sequencing errors are going to be very low in these assembled sequences, but PCR errors will remain in them. I think the authors should point out that the accuracy of this approach has not been thoroughly tested in this study, and there may be PCR errors in resulting reads.

·

Basic reporting

Submission meets all requirements.

Experimental design

Generally acceptable; specific questions/comments included below.

Validity of the findings

I am concerned about the relatively small amount of data analyzed. It is more a proof of concept than a thorough evaluation. However, this is not a sufficient reason to reject.

Additional comments

The authors repeatedly describe their method as a way to sequence full length 16s. It may be quibbling, but this is not true. However accurately, they are assembling full length 16s from shorter reads.

"We present a method to sequence full-length 16S amplicons using Illumina technology." "We have presented a method for sequencing full-length 16S rRNA gene sequences on the high throughput Illumina MiSeq platform. " Not so.

"This method enables the generation of large numbers of high quality, full-length 16S rRNA gene sequences..." This statement is accurate.


I also really dislike their choice of designations: Long-Read, Long-V4, and Long-V1-V4.
I suggest instead: Full-length, pseudo-V4, and pseudo-V1-V4. "Long-read" is misleading (the reads are 2 x 250) and "long-V4" implies a molecule longer than the standard V4 amplicon. "V4" will be generally understood to be an amplicon generated using primers that span that region, i.e., 515F/806R variants. Also "short sequencing"--V4 amplicons were sequenced using a 500 cycle v2 kit (2x250) as were the full length and tagmented templates. Sequencing presumably took the same amount of time.

If not pseudo, maybe in silico, or extracted.

Methods:

The ability to generate additional long sequences depends upon whether the primers are universal. Reference sequences are likely to cover the 1391R region, so it can be evaluated, but far less likely to include 27F except when that primer has been used in amplification. Authors should discuss this.

Why was the V1-V4 region selected as an additional comparison? V3-V6 could have been tested as well. Many reference sequences do not include the V1.

Is it FastTree2 that reports the number of resolved branches? Clarify.

All PCR used Qiagen Taq. Authors--comment on how the results might differ using a high-fidelity enzyme. I hope that most people who are sequencing 16s amplicons on next-gen platforms are using high-fidelity polymerases to construct the libraries.

Authors should indicate what fraction of the 1st PCR reaction or pool (line 101) the 31 ul represents.

Long amplicons (given the insert was >1300, the amplicons were very large) were combined with much smaller tagmented ones. Neither the very large size nor the mixture of very different sizes is recommended by Illumina. Were there any technical difficulties or reduction in expected yield?

There are a few places where a citation is lacking for a method. Phred, BWA, samtools, etc.

What is the reference database used for UCHIME? version or date of download?

PhyloSift method--I could not find a description of its demultiplexing capability in the cited paper or on the WordPress site. Is there a more recent program version that needs to be cited? https://phylosift.wordpress.com/tutorials/running-phylosift/multiplexed-samples/ implies that demultiplexing needs to be done before using PhyloSift.


Results:
What is the authors' explanation for the assembly of 'Long-Read" sequences that are only 400 or so in length?

I am curious as to what fraction of the UCHIME called chimeras were NOT found by your method--do they really appear chimeric?

Line 406 in the Results text says 88.4 %, while S2 has 83.7%.


Discussion:
This was a very small dataset--Miseq Nano--but the authors make projections up to the level of a Hiseq2500 run. Why such a small run and do they feel their conclusions are well supported?

Line 442: "...or tags too short (Lundberg et al 2013) to permit the assembly and chimera filtering techniques proposed here." Lundberg tags were 8nt vs. 10nt; are these really too short for chimera detection? Or do the authors mean that the amplicons were too short for assembly, vs. the tags?

Line 452 discusses the low reagent cost per full length 16s obtained, but this ignores the much larger investment in primer synthesis, amplicon generation, and what sounds like hands-on analysis time. These would add up enormously on a Hiseq2500 run.

Eliminate redundancy in lines 493-496.


Tables/Figures:

Based on Figure 3, it seems that the input proportions of each library type (full length vs. tagmented) could be adjusted to obtain more end+end sequences. Author comment?


Figure 6: Graphs seem overly complicated for their import.

Minor editing suggestions:

Avoid unnecessary "comma-and" constructions: see
http://www.getitwriteonline.com/archive/020204whencommabfand.htm

"Data" is a plural noun--fix verb disagreements ("data is available...")

Table 2 could use vertical dividers between three sections.

Avoid word repetition such as
"Ethics approval for this study was approved by" -- change to "was given by"

"et al." (for et alia), not "et al".

In Table S2--Use a term other than "Total" for last line. The individual OTU counts don't sum to the number given as the total, presumably because a good number of them appear in multiple datasets.

Use "subjected to" rather than "subject to" for consistency.

Assess/assessed is rather overused.

Most of the Methods is written in passive voice, but not all. If you are comfortable with "our/we" statements, use them since active voice is preferable even in Methods section.

Also be careful with use of present (used in some places) vs. past.

Line 238--period needed.

No need for italics when listing phyla.

Figure 4: "dotes" misspelled.

(This is not a comprehensive list of minor corrections.)

Reviewer 3 ·

Basic reporting

No Comments

Experimental design

No Comments

Validity of the findings

No Comments

Additional comments

Burke and Darling present an interesting approach to generate near full-length 16S rRNA gene sequences using the Illumina platform. While complex and quite difficult to follow in places (the methodology section took a few reads) I thought that this was generally an enjoyable manuscript.

If I am being honest I suspect that the complexity, low yield (and resulting cost implications) of the technique they describe will limit wide-spread adoption but most of the results and interpretation seem sound, and I had only relatively minor comments on the content. These are listed below in the order they appeared in the text:

1. Line 15 – It says here that 1612 near full length sequences were generated, but Table S2 seems to show that 1604 were generated, which is correct?

2. Lines 114 to 119 – The random tag approach is interesting. It was not clear from this section though where the primers were sourced form, or how the unique nature of the 10 nt random sections in each and every primer was assured. More information on this would be helpful. Some indication of cost might be helpful too?

3. Lines 119 to 174 – It would be helpful to state how many PCR cycles were used in total during the process of generating the tagmented full length 16S rRNA gene libraries. It looked like it was 1 + 1 + 34 + Nextera (12?), which would make perhaps 48 PCR cycles in total? This is quite a high number, with the potential to bias results? It is usually considered ideal to keep PCR cycle numbers as low as possible. From looking at Figure 7 it looks like there has not been a huge impact of differential PCR cycle numbers (I think the v4 comparative samples had just 30 cycles) in this case, but it is definitely still an issue to consider and perhaps comment on in the text.

4. Lines 212 to 217 – I wondered if this section was necessary? If you can’t assign these reconstructed 16S rRNA gene sequences to a particular sample then their usefulness is severely limited?

5. Line 347 (and Figure 7) – Enhydrobacter, Acinetobacter and Corynebacterium have all been shown to be potential kit/reagent contaminants (see Table 1 in Salter et al (2014) http://bmcbiol.biomedcentral.com/articles/10.1186/s12915-014-0087-z) and the microbial biomass recovered using skin swabs may be relatively low. Did the authors sequence any negative controls to monitor the background contamination? I suppose that even if they were contaminants it would not really impact the main message here, which is to demonstrate the 16S rRNA gene assembly approach, but it may be worth clarifying this point somewhere in the Methods section?

6. Lines 406 to 414 (and also lines 21 to 22) – The Discussion section is generally good, but is perhaps a little over-optimistic in places, and does not really highlight the disadvantages associated with their approach in comparison to the standard short-read protocols that are currently used. For balance it would be good to highlight these in more detail.

For example, on lines 406 to 407 the authors are “assuming linear scaling”. Is this assumption justified? They have no data to support this, and there are plausible theoretical reasons to suggest that scaling might not be linear (for example the shorter read lengths generated by the HiSeq might reduce the number of assembled full length sequences, or the increased error rates towards the terminal ends when using the v3 MiSeq chemistry might impact the accuracy of the assemblies?).

Similarly, the Discussion shows that the cost would be much lower per read than using traditional Sanger sequencing. This is fair enough, but since no-one is really using Sanger sequencing anymore this is not the convincing cost argument that needs to be made. It would be much better to present an estimate of the cost multiplier to achieve, say, 10,000 reads per sample, which can easily be generated currently on a single MiSeq run incorporating 200 to 300 samples. Since the proof of concept data in Table S3 shows that their new approach is comparatively much less efficient (i.e. the yield was poor, and it commonly generated only single or double-digit numbers of full-length reads) the number of samples per run, even allowing for greatly improved throughput on a HiSeq, would presumably have to be drastically reduced to achieve comparable read depth per sample?

I suspect that many will consider the reduced phylogenetic resolution you get with short-read approaches to be worth it for the considerable cost savings, which will in turn mean that many more samples can be included per run, greatly improving the power of statistical comparisons between study cohorts.

There is also the fact that the two different approaches actually appear to give you very similar results at both the phylum and genus levels (see Figure 7). So, while full length sequences are certainly desirable, there are compelling reasons why you might not want to adopt this new method.

All of these issues should ideally be included in the Discussion section in order to balance it out a little better.

7. Lines 418 to 419 (and 23 to 24) – I do agree that it could be a useful approach to provide reference sequences for environments or taxa that are poorly represented in databases. There is one caveat though, which is that poorly represented taxa tend to be low abundance organisms, otherwise they would likely have been picked up already by previous low-throughput Sanger sequencing-based studies and full-length 16S rRNA gene reference sequences would be available? The rather low yield of their approach could conceivably limit full-length sequences to predominantly abundant taxa? This should probably be considered in the text.

8. Lines 648 to 649 – Readers may not be familiar with “the standard Nextera method” so it is probably better to include a brief recap of that approach in the relevant Methods section instead, perhaps at line 160?

9. Line 676 – “dotes” should be “dots”?

Minor pedantic points regarding terminology:

10. The word microbiome is not really appropriate in the context of community profiling approaches such as 16S rRNA gene sequencing since this technique just describes the composition of a microbial community rather than the wider environmental habitat (see a recent editorial that expands on this point here: http://microbiomejournal.biomedcentral.com/articles/10.1186/s40168-015-0094-5). As such all uses of “microbiome” should really be replaced with “microbiota”.

11. Similarly, the multiple uses of “16S” throughout the manuscript (including figures/legends) should really be replaced with “16S rRNA gene” as there is no such thing as “16S”.

12. Finally, the full length of the 16S rRNA gene is closer to 1500 bp, rather than the 1372 bp fragments that were assembled in the current paper. As a result “full-length” should probably be replaced throughout the manuscript by “near full-length”.

---

## Round 0.2 · Minor Revisions

Before Acceptance, there is just a couple of final minor comments from Reviewer 3.

·

Basic reporting

The authors addressed all my previous concerns.

Experimental design

The authors addressed all my previous concerns.

Validity of the findings

The authors addressed all my previous concerns.

·

Basic reporting

Submission meets all requirements.

Experimental design

Submission meets all requirements.

Validity of the findings

Submission meets all requirements. I appreciate that the authors have specified that the work is proof of principle.

Additional comments

I consider the revisions adequate to address my earlier criticisms and those of the other reviewers. The authors have made an acceptable case for describing the procedure as 'near full-length sequencing' vs. assembly.

Reviewer 3 ·

Basic reporting

No comments

Experimental design

No comments

Validity of the findings

No comments

Additional comments

The authors' response largely addresses my previous minor concerns.

I would only suggest that they add a part of the text from the response letter "Bioanalyser traces showed no evidence of a PCR product, although it is possible that products below the limit of detection may still have been present." to line 160 (note that this is using the line numbering from the PDF version).

Similarly there were a few remaining erroneous uses of "16S" (see lines 69, 70, 102, 111, 279 in the PDF version, and Figures 1 and 2 plus legends), and one use of "microbiome" that should be replaced with "microbiota" on line 453.

---

## Round 0.3 · accepted · Accept

We are pleased to inform you that after two rounds of review assessments, this manuscript is now accepted for publication.